# AN ACTIVE LEARNING FRAMEWORK FOR EFFICIENT ROBUST POLICY SEARCH

## ABSTRACT

Robust Policy Search is the problem of learning policies that do not degrade in performance when subject to unseen environment model parameters . It is particularly relevant for transferring policies learned in a simulation environment to the real world. Several existing approaches involve sampling large batches of trajectories which reflect the differences in various possible environments, and then selecting some subset of these to learn robust policies, such as the ones that result in the worst performance. We propose an active learning based framework, EffAcTS, to selectively choose model parameters for this purpose so as to collect only as much data as necessary to select such a subset. We apply this framework to an existing method, namely EPOpt, and experimentally validate the gains in sample efficiency and the performance of our approach on standard continuous control tasks. We also present a Multi-Task Learning perspective to the problem of Robust Policy Search, and draw connections from our proposed framework to existing work on Multi-Task Learning.

## 1 INTRODUCTION

Recent advances in Deep Reinforcement Learning (DRL) algorithms have achieved remarkable performance on continuous control tasks (Schulman et al. (2015); Lillicrap et al. (2016); Schulman et al. (2017); Wang et al. (2017)). These model-free algorithms, however, require an abundance of data, severely restricting their utility on non-simulated domains. Thus, when it comes to "real" domains such as physical robots, model-based approaches such as (Levine & Koltun (2013); Deisenroth & Rasmussen (2011)) have been favored for their significantly better sample complexity. One class of model based approaches that has gained traction is to learn from multiple simulated domains that approximate the real "target" domain. These usually correspond to an ensemble of models with various parameters such as the mass of a part of a robot or the coefficient of friction between the robot's foot and the ground. The problem of *robust policy search* is to learn policies that perform well across this ensemble.

One prominent class of approaches toward this goal involves sampling model parameters from the ensemble and collecting batches of trajectories at these parameters (Wang et al. (2010); Rajeswaran et al. (2017); Yu et al. (2017)). These approaches differ mainly in the way in which they choose subsets of these trajectories to focus on for policy learning. Although robust policy search is inevitably a harder learning problem than standard policy search, amount of data collected by these methods is still quite large, up to almost 2 orders of magnitude more than is usually required by the usual policy optimization algorithms, regardless of the method used to choose a subset of these trajectories for learning (and thus possibly discarding a large portion of the data).

We introduce an active learning framework for intelligently selecting model parameters for which to sample trajectories so as to directly acquire some desired subset of the trajectories (such as the subset resulting in the worst performance), while collecting as little additional data as possible. We then present an instantiation of the framework in the setting of an existing robust policy search approach, EPOpt, and perform experimental validation on environments from standard continuous control benchmarks to empirically demonstrate significant reductions in sample complexity while still being able to learn robust policies. We also explore connections to Multi-Task Learning that are revealed upon casting Robust Policy Search as a Multi-Task Learning problem and discuss its relation to existing work in the area.

## 2 RELATED WORK

Wang et al. (2010) learn controllers with a specific funtional form using trajectories sampled for parameters drawn from an ensemble, and optimize for the average case performance. Rajeswaran et al. (2017) propose EPOpt, which learns a Neural Network (NN) policy using a model-free DRL algorithm, but on simulated domains sampled from an ensemble of models. An adversarial approach to training is taken that involves selectively exposing to the model-free learner only data from those sampled models on which it learner exhibits the least performance. Even though this is a more sophisticated approach than the former and is demonstrated to have greater performance and robustness, the number of trajectories collected is still very large. Yu et al. (2017) also propose an approach that optimizes the average case performance, but additionally performs explicit system identification, and the estimated model parameters are fed to a NN policy as additional context information alongside the original observations. Again, the data requirements are quite large, both for policy learning as well as system identification. Approaches related to learning from an ensemble of models have also been studied under Domain Randomization (Tobin et al. (2017)).

A recent work that learns from an ensemble of models is (Kurutach et al. (2018)), but the ensemble here consists of learned DNN models of the dynamics for use in Model Based RL, rather than being induced by changing physical properties of the environment. A similar ensemble generated by perturbing an already learned model is used for planning through in (Mordatch et al. (2015)). This work also does not deal with model uncertainties with physical meaning.

Although EPOpt uses only an appropriate subset of models to train on, none of the above approaches consider ways to sample trajectories only as necessary. Our proposed framework employs active learning to decide with data from only a few model parameters the models for which the agent requires more training. Active sampling approaches have also been explored for task selection in Multi-Task learning by Sharma et al. (2018), a viewpoint we discuss in more detail in section 5.

## 3 BACKGROUND

### 3.1 RL ON AN ENSEMBLE OF MODELS

We work with the same setting described in (Rajeswaran et al. (2017)) where the model ensemble is represented as a family of parametrized MDPs on a fixed state and action space. Following the same notation, this is the set $\mathcal{M}(p) = \langle \mathcal{S}, \mathcal{A}, \mathcal{T}_p, \mathcal{R}_p, \gamma, \mathcal{S}_{0,p} \rangle$ for each parameter $p$ in the space of parameters $\mathbb{P}$, whose elements are respectively the state and action spaces, transition functions, reward functions, discount factor and the initial state distribution. Those items that are subscripted with $p$ depend on $p$, i.e different parameters induce different dynamics and rewards. We note here that we say "parameter" even if it is a vector rather than a real number. Further, there is a source distribution $\mathcal{P}$ that indicates the likelihood of any particular $p \in \mathbb{P}$ in the model ensemble.

We denote the typical trajectory from any of these MDPs by $\tau = \{s_t, a_t, r_t\}_{t=0}^{T}$, where $T$ is the time horizon, and the discounted return from the start state $R(\tau) = \sum_{t=0}^{T} \gamma^t r_t$. These trajectories are generated by following a policy which are parameterized by a vector $\theta$, which we denote by $\pi_\theta$. We define the performance at parameter $p$ as the expected discounted return from the start state in $\mathcal{M}(p)$

$$\eta(\theta, p) = \mathbb{E}_\tau \left[ \sum_{t=0}^{T} \gamma^t r_t \,\middle|\, p \right]$$

### 3.2 ROBUST POLICY SEARCH AND EPOPT

Robust Policy Search seeks policies that perform well across all parameters in $\mathbb{P}$, and do so without knowing the parameter for the MDP on which they are being tested. This translates to being able to perform well on some unknown target domain, and also potentially handle variations not accounted for in $\mathbb{P}$. The intuitive objective for this is to consider the average performance of the policy over the source distribution $\eta_D(\theta) = \mathbb{E}_{p \sim \mathcal{P}}[\eta(\theta, p)]$. However, this objective could be close to the maximum even if there are sharp dips in some regions of $\mathbb{P}$, which is why EPOpt uses instead

---

**Algorithm 1** EPOpt: Trajectory generation and outer loop

| | |
|---|---|
| 1: **function** LEARN($N_{iters}$, $N$, $\epsilon$, $\theta_0$) | 1: **function** |
| 2:      **for** $i = 0 \dots N_{iters} - 1$ **do** |      GETTRAJECTORIES($N$, $\epsilon$, $\theta_0$) |
| 3:          $\boldsymbol{\tau}_C \leftarrow$ GetTrajectories($N$, $\epsilon$, $\theta_i$) | 2:      $P \leftarrow$ Sample of $N$ model parameters from |
| 4:          $\theta_{i+1} \leftarrow$ BatchPolOpt($\boldsymbol{\tau}_C$, $\theta_i$) |      $\mathcal{P}$ |
| 5:      **end for** | 3:      $\boldsymbol{\tau} \leftarrow$ Trajectories collected for each $p \in P$ |
| 6:      **return** $\theta_{N_{iters}}$ | 4:      $\boldsymbol{\tau}_C \leftarrow$ Subset of $\boldsymbol{\tau}$ correspondig to the bottom $\epsilon$ percentile of returns |
| 7: **end function** | 5:      Return $\boldsymbol{\tau}_C$ |
| | 6: **end function** |

---

the Conditional Value at Risk (CVaR) formulation from (Tamar et al. (2015)), which considers the performance across only the subset of $\mathbb{P}$ that corresponds to the bottom $\epsilon$ percentile of returns from $\mathcal{P}$. This has the effect that policies which have such sharp dips (i.e bad worst case performance) are no longer considered good solutions.

EPOpt approximately samples trajectories for parameters from this subset of $\mathbb{P}$ by first sampling $N$ trajectories from the entire source distribution, finding the discounted returns for these and then taking the bottom $\epsilon$ percentile according to the evaluated returns. These trajectories are then sent to a standard batch policy optimization algorithm (such as Trust Region Policy Optimization (TRPO) (Schulman et al. (2015))) denoted as BatchPolOpt and processed as usual to update the policy parameters $\theta$. EPOpt is summarized in Algorithm 1.

### 3.3 LINEAR STOCHASTIC BANDITS

Here, we provide a quick overview of Linear Stochastic Bandits (LSB) since they play an important role as solutions to the Active Learning problem in section 4.

The LSB problem is one of finding the optimal arm from a given set of arms $\mathcal{X}$ similar to the standard Multi-Armed Bandit (MAB) problem, but with the average reward from each arm being an unknown linear function of the features associated with that arm. That is, if $x \in \mathcal{X}$ is an arm, and we also denote its features by $x$, the reward is given by $r(x) = x^T \theta^* + \xi$, where $\xi$ is some zero-mean noise, and $\theta^*$ gives the parameters for said linear function. Thus, finding the optimal arm amounts to estimating $\theta^*$.

There have been several approaches to solving the LSB problem under various objectives. One group of works (Abbasi-Yadkori et al. (2011); Li et al. (2010)) are based on the principles of the Upper Confidence Bound Algorithm for MAB problems. The other popular class of approaches (Abeille & Lazaric (2017); Agrawal & Goyal (2013)) is based on Thompson Sampling.

## 4 ACTIVE LEARNING FOR EFFICIENT TRAJECTORY SAMPLING

To motivate our developments to improve on the sample efficiency, we start with the observation that there is some functional dependence of the performance of a given policy on the model parameter corresponding to a task under consideration. The CVaR objective estimation process of EPOpt disregards this functional dependence as it does not assume such a thing exists (in the sense that even if the performance was completely independent of the model parameter, it would still be able to come up with a sample from the bottom $\epsilon$ percentile of trajectories). Thus, an unavoidable side effect of this process is that EPOpt discards many of the trajectories it collects (for $\epsilon = 0.1$, 90% of the collected trajectories are discarded). Here, we wish to devise a strategy to utilize this information effectively so as to minimize such wastage.

Active Learning is a paradigm where the agent chooses data to learn from based on its previous experience (see (Settles (2009)) for a comprehensive survey). It has been used to speedup learning tasks, especially in situations with limited data. An active learner not only needs to work with as few samples as possible, it also needs to account for the uncertainty in whatever data it has collected. These are exactly the desiderata of the required strategy, as it must be able to fit the performance function across $\mathbb{P}$ by collecting as few trajectories as possible, which come with noisy evaluations

---

**Algorithm 2** Trajectory generator for EffAcTS-EPOpt.

1: **function** GETTRAJECTORIES($N_C$, $N_B$, $\epsilon$, $B$)
2:     **for** $i = 1 \dots N_B$ **do**
3:         Sample parameter $p_i$ as an optimal arm estimate by $B$.
4:         Collect a trajectory $\tau_i$ at parameter $p_i$ and find the return $R_i$.
5:         Update $B$ with $(\tau_i, -R_i)$
6:     **end for**
7:     $\mathbf{P} \leftarrow \{\}, \hat{\mathbf{R}} \leftarrow \{\}$         ▷ Buffers to store parameters and corresponding return estimates
8:     **for** $j = 1 \dots \left\lceil \frac{N_C}{\epsilon} \right\rceil$ **do**
9:         Sample parameter $p_j$ from source distribution
10:         $\hat{R}_j \leftarrow$ Return estimate by $B$ at $p_j$
11:         Add $\hat{R}_j$ to $\hat{\mathbf{R}}$ and $p_j$ to $\mathbf{P}$
12:     **end for**
13:     $\mathbf{P}_C \leftarrow$ Subset of $\mathbf{P}$ giving rise to the bottom $\epsilon$ percentile of returns in $\hat{\mathbf{R}}$
14:     $\tau_C \leftarrow$ Trajectories collected for each $p \in \mathbf{P}_C$
15:     **return** $\tau_C$
16: **end function**

---

of the performance. Thus, quite clearly, the problem of efficiently performing such sampling is connected to active learning. The case of active learning that is of interest to us is when the agent is allowed to sample output for arbitrary points in the input space (instead of having to choose points from a finite dataset). The input here is some parameter $p$ from $\mathbb{P}$, and the output is the return from one trajectory collected at $p$.

We now outline our active learning framework to selectively generate the trajectories. In this procedure, the active learner sequentially picks some parameters and trajectories are sampled for each of them. After a particular number of trials, the learner in theory is expected to have a reasonably good fit to the performance as a function of the parameters. This can then be used to decide which parts of $\mathcal{P}$ trajectories need to be collected for according to some given objective. We call the resulting framework EffAcTS (Efficient Active Trajectory Sampling). Any particular instantiation of EffAcTS is defined by the choice of active learner and also the scheme used to select which parts of $\mathcal{P}$ to sample from.

We analyze one such instantiation based on EPOpt. The trajectory selection scheme is the same as in EPOpt, to generate samples of the bottom $\epsilon$ percentile of trajectories. The difference is that this is done by sampling a batch of parameters from $\mathcal{P}$, and collecting trajectories for only those that are in the worst $\epsilon$ percentile according to the function fit by the active learner. Bandit algorithms are natural choices for active learning as they directly satisfy the aforementioned requirements, and are well studied in both Multi-Armed Bandit (Antos et al. (2008); Carpentier et al. (2011)) as well as Linear Stochastic Bandit (Soare et al. (2013)) settings. Since the parameter space is invariably continuous, we turn to LSBs as the active learner in our experiments. Feedback is given to it in an adversarial manner, being proportional to the negative of the return obtained on that sampled trajectory. This causes it to seek out regions with low performance, and in the process learn about the performance across $\mathcal{P}$. We note that although the bandit's learning phase is inherently serial, it is still possible to collect the trajectories for the estimated worst $\epsilon$ percentile of parameters in parallel.

We call this algorithm EffAcTS-EPOpt and the above trajectory generation routine is described in Algorithm 2. This routine is given as (fixed) input the following quantities: $N_B$, the number of trajectories sampled by the bandit in the course of its learning, $N_C$, the total number of trajectories collected as a sample from the worst $\epsilon$ percentile of trajectories (this means that $\left\lceil \frac{N_C}{\epsilon} \right\rceil$ parameter values are drawn from the source distribution and their performance is estimated using the bandit, but only the bottom $N_C$ of those are used to collect trajectories). The most critical component is the LSB learner $B$ which is sequentially fed parameter values and the negative of the return obtained on one sampled trajectory at each parameter. It also incorporates internally a feature transformer $f$ that takes in a parameter from the source distribution's support and applies some transformation on it (including possibly standardization), and also scales the negative returns given to it appropriately.

Clearly, due to the fact that EffAcTS-EPOpt does not need to discard a $1 - \epsilon$ fraction of the trajectories it collects, it is much more sample efficient. If EPOpt collects $N$ trajectories, and EffAcTS-

EPOpt's bandit learner is allowed $N_B$ "arm pulls", with the same number of trajectories being collected for the bottom $\epsilon$ percentile as EPOpt (i.e $N_C = \epsilon N$), the ratio of the total amount of data collected is $\left(\frac{N_B + \epsilon N}{N}\right)$. For a nominal setting of $N = 240$, $\epsilon = 0.1$ and $N_B = 24$, this results in a dramatic 80% reduction in the amount of data collected.

## 5 CONNECTIONS TO MULTI-TASK LEARNING

The problem of robust policy search on an ensemble of models can also be viewed as a form of transfer learning from simulated domains to an unseen real domain (possibly without any training on the real domain, which is referred to as direct-transfer or jumpstart (Taylor & Stone (2009))). Further, the process of learning from an ensemble of models can be viewed as a Multi-Task Learning (MTL) problem with the set of tasks corresponding to the set of parameters that constitute the source domain distribution. Learning a robust policy corresponds to maintaining performance across this entire set of tasks, as is usually the goal in MTL settings. MTL, which is closely related to transfer learning, has been studied in the DRL context in a number of recent works (Rusu et al. (2016a;b); Parisotto et al. (2016); Sharma et al. (2018)). However, these works consider only discrete and finite task sets, whereas model parameters form a (usually multi-dimensional) continuum. More generally, we can think of MTL with the task set being generated by a set of parameters, and refer to such problems as *parameterized MTL* problems, with Robust Policy Search being an instance of this setting.

(Sharma et al. (2018)) has employed a bandit based active sampling approach similar to what we have described here to intelligently sample tasks (from a discrete and finite task set) to train on for each iteration. The feedback to the bandit is also given in an adversarial manner. However, we note several differences when it comes to a parameterized MTL setting. The first is the functional dependence of the task performance to an underlying parameter as discussed earlier. In the discrete MTL settings usually studied (such as Atari Game playing tasks), there is no such visible dependency that can be modeled. This means that any algorithms that are adapted to the parameterized setting need to be reworked to utilize such dependencies as EffAcTS has done. The task selection procedure in EffAcTS differs from the one in (Sharma et al. (2018)) in that the performance is sampled for several tasks in between iterations of policy training. Additionally, one single task is not chosen in the end, rather the bandit's learning is used to inform the selection of a group of tasks as necessary for the CVaR objective.

## 6 EXPERIMENTS

We conduct experiments to answer the following questions to analyze EffAcTS-EPOpt:

- Do the policies learned using EffAcTS-EPOpt suffer any degradation in performance from that of EPOpt? Is robustness preserved across the same range of model parameters as in EPOpt?

- Does the bandit active sampler identify with reasonable accuracy the region corresponding to the worst $\epsilon$ percentile of performance, and does it achieve a reasonable fit to the performance across the range of parameters (i.e has it explored enough to avoid errors due to the noisy evaluations it receives)?

- How much can the sample efficiency be improved upon? This mainly boils down to asking how few trajectories are sufficient for the bandit to learn well enough.

The experiments are performed on the standard Hopper and Half-Cheetah continuous control tasks available in OpenAI Gym (Brockman et al. (2016)), simulated with the MuJoCo Physics simulator (Todorov et al. (2012)). As in (Rajeswaran et al. (2017)), some subset of the following parameters of the robot can be varied: Torso Mass, Friction with the ground, Foot Joint Damping and Joint Inertias. We also use the same statistics for the source distributions of the parameters which are described in Table 1.

For all our experiments, we implement the bandit learner using Thompson Sampling due to its simplicity. In order to allow for some degree of expressiveness for the fit, we apply polynomial

| Parameter | $\mu$ | $\sigma$ | Low | High | Parameter | $\mu$ | $\sigma$ | Low | High |
|---|---|---|---|---|---|---|---|---|---|
| Mass | 6.0 | 1.5 | 3.0 | 9.0 | Mass | 6.0 | 1.5 | 3.0 | 9.0 |
| Friction | 2.0 | 0.25 | 1.5 | 2.5 | Friction | 0.5 | 0.1 | 0.3 | 0.7 |
| Damping | 2.5 | 2.0 | 1.0 | 4.0 | Damping | 1.5 | 0.5 | 0.5 | 2.5 |
| Inertias | 1.0 | 0.25 | 0.5 | 1.5 | Inertias | 0.125 | 0.04 | 0.05 | 0.2 |

Table 1: Description of the source domain distribution for the Hopper (Left) and Half-Cheetah (Right) tasks. The values given here specify the means ($\mu$) and standard deviations ($\sigma$) for normal distributions truncated at the low and high points mentioned here. This is equivalent to the probability density of a normal distribution with parameters ($\mu, \sigma$) being zeroed outside the interval [Low, High], and being normalized so that it integrates to 1.

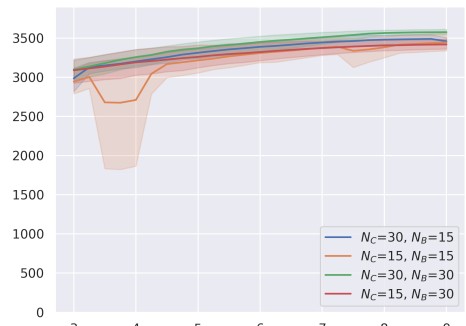 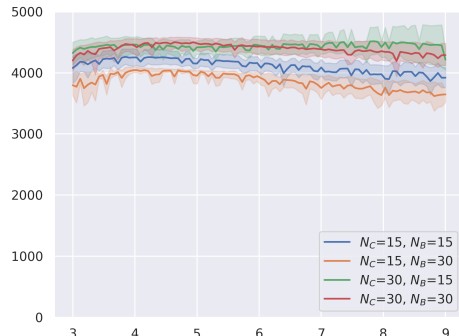

Figure 1: Performance as a function of torso mass for $\epsilon = 0.1$ for the Hopper (Left) and Half-Cheetah (Right) tasks. The bands indicate the confidence intervals of the performance measured across 5 runs of the entire training procedure.

transformations of some particular degree to the model parameters. Policies are parameterized with NNs and have two hidden layers with 64 units each, and use tanh as the activation function.

We emphasize that because we are using the same environments in (Rajeswaran et al. (2017)) and also the same policy parameterization, results reported there can be used directly to compare against EffAcTS-EPOpt.

### 6.1 PERFORMANCE AND ROBUSTNESS

First, the torso mass is varied in both the Hopper and the Half-Cheetah domains keeping the rest of the parameters fixed at their mean values. The performance of the EffAcTS-EPOpt learned policy is then tested across this range, and the results are shown in Figure 1. We use Trust Region Policy Optimization (TRPO) (Schulman et al. (2015)) for batch policy optimization and run it for 150 iterations. For this part, we use 4th degree polynomial transformations.

We see that the policy is indeed robust as it maintains its performance across the range of values of the torso mass, and it achieves near or better than the best performance for both tasks as reported in (Rajeswaran et al. (2017)) in all but one case, with $N_C = 15$ and $N_B = 15$ for Hopper. This setting samples the least number of trajectories per iteration, 30, which is just one eighth of the 240 drawn in EPOpt. Although there is one region where it is unstable, it is still able to maintain its performance everywhere else.

For the other settings which use more trajectories, this does not happen, and even at $N_C = 30$ and $N_B = 30$ which samples the most trajectories, a 75% reduction in samples collected is achieved over EPOpt (the total number of iterations is the same). The other two settings which perform almost as well in both tasks collect 45 each, which amounts to an even larger reduction of 81.2%. In the case of $N_C = 15$ and $N_B = 15$ with the Half-Cheetah Task, this number is pushed even further to 87.5% while still retaining performance and robustness.

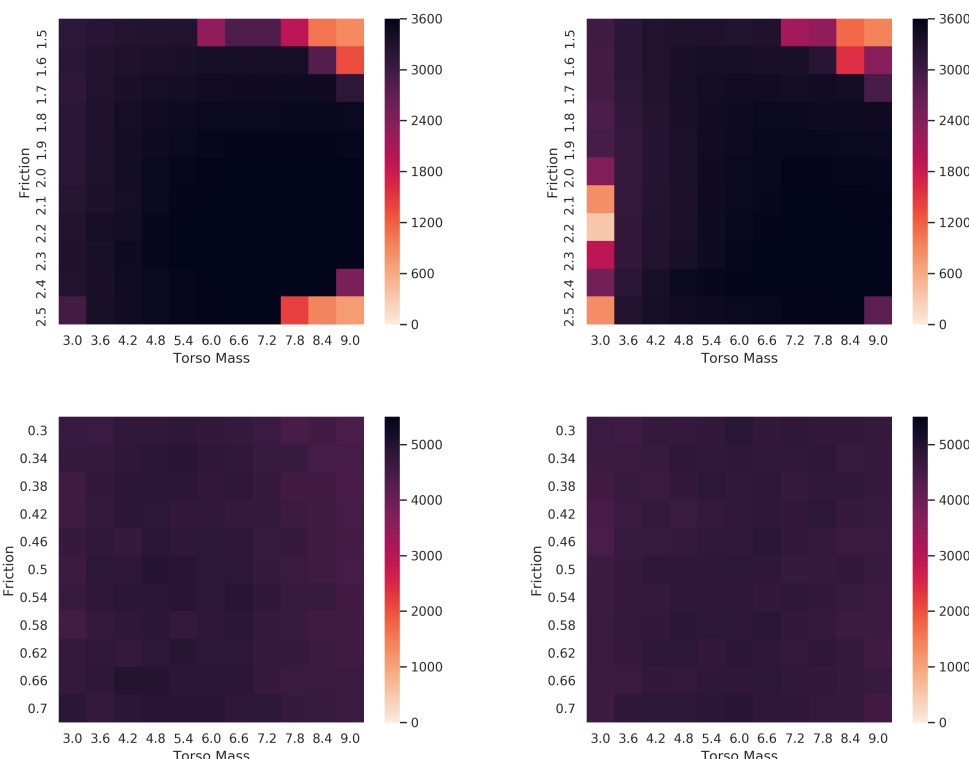

Figure 2: Performance as a function of torso mass and ground friction on the Hopper (Top row) and Half-Cheetah (Bottom row) tasks for $\epsilon = 0.1$ for one run or EffAcTS-EPOpt. In each row, Left: $N_C$=30, $N_B$=30. Right: $N_C$=30, $N_B$=50.

Next, in both the tasks, we vary the friction with the ground in addition to the torso mass, thus creating a two dimensional ensemble of parameters. Here, we run TRPO for 200 iterations, and again use 4th degree polynomial transformations. Figure 2 shows the results obtained.

Full performance is maintained over almost all of the parameter space in both domains, again being comparable to or better than in (Rajeswaran et al. (2017)). Notably, with $N_C$=30, $N_B$=30, the same 75% reduction in collected trajectories is obtained even in a higher dimensional model ensemble. This is also despite the added challenge of an increased number of parameters for the bandit to fit (15 as opposed to 5 in the previous experiment).

## 6.2 ANALYSIS OF BANDIT ACTIVE LEARNER

Here we validate one of our key assumptions, that the active learner learns well enough about the performance that it can produce a decent approximation of a sample batch of the bottom $\epsilon$ percentile of trajectories. For this, we first evaluate the average return for a batch of samples from $\mathcal{P}$ by collecting a large number of trajectories at each parameter. We note that these trajectories are solely for the purpose of analysis and are not used to perform any learning. Then, the percentile of the trajectory deemed to have the greatest return among those chosen based on the bandit learner is computed using the returns in this batch by using a nearest-neighbor approximation. This is done across several iterations during the training, and the median percentiles along with other statistics are reported in Table 2 for the Hopper task.

Ideally, we would like this value to come out to around $100\epsilon$ (when written as proper percentiles), i.e the parameter with the greatest performance among the worst $\epsilon$ percentile should be at the $\epsilon^{th}$

| Hyperparameters | Median %tile | Avg %tile | Std. Dev. | Max %tile |
|---|---|---|---|---|
| $N_C$=15, $N_B$=15 | 14.6 | 15.9 | 7.7 | 32.2 |
| $N_C$=15, $N_B$=30 | 14.9 | 23.4 | 24.0 | 98.0 |
| $N_C$=30, $N_B$=15 | 13.7 | 23.0 | 19.4 | 74.7 |
| $N_C$=30, $N_B$=30 | 11.6 | 14.9 | 10.6 | 47.8 |

Table 2: Statistics for the percentiles described in section 6.2 for the Hopper task's 1-D model ensemble with $\epsilon = 0.1$, which are measured every fifth iteration from the 100th to the 150th iterations.

percentile. In our estimates, there are some outliers that cause the average to become large, but as the median value shows, it is indeed reasonably close to the desired value of 10 for $\epsilon = 0.1$.

## 7 CONCLUSIONS AND FURTHER POSSIBILITIES

We developed the EffAcTS framework for using active learning to make an informed selection of model parameters based on agent performance, which can subsequently be used to judiciously generate trajectories for robust RL. With an illustration of this framework based on EPOpt, we have both demonstrated its applicability for robust policy search as well as established its effectiveness in reducing sample complexity by way of empirical evaluations on standard continuous control domains. We also discussed our work in the context of Multi-Task Learning along with the similarities and differences between these settings.

Our work opens up requirements for active learning algorithms that can work well with even lesser data than the ones tested here. Methods like Gaussian Process Regression are known to be efficient, but not in high dimensional spaces. For robust policy search methods to be effective for transfer from simulation to reality, they need to be able to handle the complexities of the real world, which necessitates methods that work with high dimensional model ensembles, which in turn entail frameworks such as EffAcTS to help reduce the sample complexity. Another possibility for robust policy search itself is to develop objectives that can speedup learning as well as make use of the features of EffAcTS to maintain sample efficiency.

With our interpretation of robust policy search as a parameterized version of Multi-Task Learning, a natural next step would be to adapt developments in the usual discrete MTL setting to robust policy search. It would also be worthwhile to similarly investigate the applicability of Meta Learning, as it would prove useful for both dealing with large disparities between the source domains and the real world, as well as coping with unmodeled dynamics (which are unavoidable since it is not feasible to model the real world with complete accuracy).

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

## A   APPENDIX

### A.1   IMPLEMENTATION NOTES

We use the following hyperparameter settings for TRPO suggested in OpenAI Baselines Dhariwal et al. (2017) on which our implementation is also based:

| Hyperparameter | Value |
|---|---|
| Timesteps per batch | 1024 |
| Max KL | 0.01 |
| CG Iters | 10 |
| CG Damping | 0.1 |
| $\gamma$ | 0.99 |
| $\lambda$ | 0.98 |
| VF Iterations | 5 |
| VF Stepsize | 1e-3 |

Table 3: Hyperparameters for TRPO

An important difference from EPOpt's implementation is that we use Generalized Advantage Estimation Schulman et al. (2016) instead of subtracting a baseline from the value function. For value function estimation using a critic, we use the same NN architecture as the policy, 2 hidden layers of 64 units each.

For our bandit learner, we used Thompson Sampling and followed the version described in Abeille & Lazaric (2017). The hyperparameters introduced by this are as follows (they have the same names as in the paper):

| Hyperparameter | Value |
|---|---|
| $R$ | 5.0 |
| $\delta$ | 0.1 |
| $\lambda$ | 0.5 |

Table 4: Thompson Sampling Hyperparameters

The first two parameters control the amount of exploration performed during Thompson Sampling, while $\lambda$ is a regularization parameter for the parameter estimates.

The arms of the bandit are model parameters taken uniformly across the domain and converted to feature values. The features input to the bandit are 4th degree polynomial terms generated from the model parameters. This amounts to 5 terms in the 1-D case and 15 in the 2-D case. These arm "representations" are then standardized before being used by the bandit. The negative returns given as feedback to the bandit are scaled by a factor of $10^{-3}$.

### A.2   VISUALIZING THE BANDIT LEARNER IN EFFACTS-EPOPT

In Figure 3, we present the outcome from one particular iteration of training. The true performance profile is estimated by collecting 100 trajectories to calculate the mean return at each parameter (again, these trajectories are not used for learning). Along with this is shown the bandit's fit, the trajectories it collects while learning and the trajectories that are sampled based on its estimate of the bottom $\epsilon = 0.1$ percentile (output used for training the policy).

We see that the bandit takes exploratory actions (points to the left and in the middle) that don't lead to the worst returns. However, it quickly moves towards the region with low returns, and the final fit is close to the true mean performance. From comparison with the output trajectories, the trajectories in the learning phase are quite clearly not representative of the bottom $\epsilon = 0.1$ percentile according to the source distribution. Thus, we cannot reuse these to perform learning with the CVaR objective.

We also note that a perfectly learned performance profile is not necessary to sample from the worst trajectories. As long as the fit is reasonably accurate in that region, the output trajectories will be of

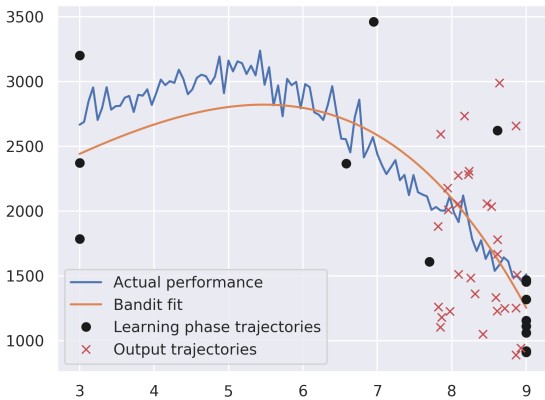

Figure 3: The bandit's operation in the 115th iteration of training in the Hopper task. Individual trajectories are shown as dots or crosses, with the position indicating the parameter value and the return obtained.

good quality. In practice, we expect LSB algorithms to be capable of doing this as they tend to focus on these regions.

### A.3 OTHER REMARKS

We note that we do not perform "pre-training" as in EPOpt where the entire batch of trajectories is used for policy learning for some iterations at the beginning (corresponding to optimizing for the average return over the ensemble). This has been reported to be necessary, possibly due to problems with initial exploration if using the CVaR objective from the beginning. EffAcTS-EPOpt on the other hand works without any such step. However, at the very beginning, we collect trajectories for parameters sampled directly from $\mathcal{P}$ until 2048 time steps have elapsed in that iteration. This is because algorithms like TRPO have been known to require at least that much data per iteration.

The bandit implementation we have is stationary, although it would still be possible to investigate non-stationary versions, where data collected from previous iterations can also be used to estimate parameters. These points would, of course be weighted down in the Linear Regression step of Thompson Sampling.

