# OpenReview forum: "An Active Learning Framework for Efficient Robust Policy Search"
_ICLR.cc/2019/Conference_

### Official Review · AnonReviewer1 · 2018-11-02
**need further improvement**

**Rating:** 5
**Confidence:** 4

**Review:**

Summary: This paper proposes an integration of active learning for multi-task learning with policy search. This integration is built on an existing framework, EPOpt, which each time samples a set of models and a set of trajectories for each model. Only trajectories with the bottom \epsilon percentile returns will be used to update the multi-task policy. This paper proposes a way to improve the sample-efficiency so that fewer trajectories will be sampled and fewer trajectories will be loss.

In general, the paper presentation is easy to follow. The idea is well motivated of why an active learning integration is needed. The related work is a bit too narrow, e.g. work [1] on the same approach like EPOpt or meta-learning (for model adaptation) [2] (and others more on this topic)

[1] T. Kurutach, I. Clavera, Y. Duan, A. Tamar, and P. Abbeel. Model-Ensemble Trust-Region Policy Opti
mization. In ICLR, 2018.

[2] C. Finn, P. Abbeel, and S. Levine. Model-Agnostic Meta-Learning for Fast Adaptation of Deep Networks. In ICML, 2017.

In overall, I have major concerns regarding to the proposed framework.
- Active learning is a method that is in general known to be an optimal trade-off between exploration vs. exploitation in finding a global optimal solution. That means, the proposed use of linear stochastic bandits is trying to find an optimal arm \theta^* (the worst trajectory) that gives the highest reward (the lowest return). In my opinion, integrating this idea naively into EPOpt to sample a set of trajectories would only aim to find the worst trajectory among all trajectories from all models. This is clearly not enough to say "finding ALL the WORSE regions among trajectory space" to improve the policy. Therefore, a new way of integration or a new objective should be used in order to make a principled framework.

- The statement over sample-efficiency gain vs. EPOpt in Section 4 is too loose which is not based on any detailed analysis or further theoretical results.

- The experiment results are not well presented: there is no results for EPOpt in Fig. 1;


Minor comments:

- Algorithm 1: argument of GetTrajectory (in LEARN) should be \theta_i, instead of \pi_\theta_i?.


In conclusion, the proposed framework is not yet principled. Experiment results are too preliminary and not well presented.

---

> ### Author Response · Authors · 2018-11-21
> **Response to review**
>
> Thank you for your review. We have made several changes to the paper as we have described in our official comment. Our response is based on this new version.
>
> Model-Ensemble Trust-Region Policy Optimization (ME-TRPO):
> This paper presents a Model Based RL method to solve any given RL task, with the highlight being that it uses an ensemble of DNNs to successfully train dynamics models for Model Based RL. The contributions of ME-TRPO do not include learning policies robust to changes in the *environment*, and it is therefore not for the same purpose as EPOpt. There is an element of dealing with model uncertainty that is shared with EPOpt, but this comes not from the environment itself but rather due to learning dynamics from finite samples. Therefore, we don’t believe a comparison with ME-TRPO is warranted. Nevertheless, we have mentioned these connections along with Ensemble CIO, a similar work.
>
> Model-Agnostic Meta-Learning:
> We wish to learn a robust policy that works ”out of the box” on a given test environment. This method on the other hand involves an adaptation phase for meta-learning for each new test task, which is not the paradigm we want. It is therefore not pertinent in the discussion on connections to Multi-Task Learning.
>
>
> Active learning vs. Explore/Exploit, Finding bottom epsilon percentile region:
> Active learning is concerned with improving and speeding up learning by making use of the ability to choose the data to learn from. The learning task in question is to perform regression from the space of env. model parameters to the performance of the policy, and it is possible to query the performance (noisily) at arbitrary parameters. The role of the active learner in our framework then boils down to (judiciously) using this ability to obtain a good fit while reducing uncertainty across its input space. So, while this involves an element of balancing exploration and exploitation, achieving an optimal trade-off is in no way the aim of the active learner.
>
> Even our use of the bandit in conjunction with EPOpt is to learn about the performance across the space of parameters (which amounts to getting a reliable estimate for the weights for its input features), and *not* for identifying the point leading to the worst performance. Although, for simplicity, we have not used active learning oriented formulations like (Soare et al. (2013)), our Thompson Sampling implementation still has good guarantees on learning accurate weights (see (Abeille & Lazaric (2017) on which it is based). With the added expressiveness of polynomial input transformations, we expect it to be able to identify accurately the bottom epsilon percentile region. Also, for the purpose of using the fit to sample from that region, it doesn’t matter if the fit is not very accurate in other regions.
>
> Now, it is entirely possible that the performance profile is too complex to fit accurately. However, as we have shown by successfully learning robust policies on even a 2-D model ensemble, this is not a problem in practice. Furthermore, even EPOpt will require too much data to correctly identify the required region given a sufficiently complex profile, and will still not be completely reliable.
>
>
> Efficiency gains:
> The statement in Section 4 calculates as an illustrative example the reduction in data requirement from EPOpt under a particular value of trajectory allowances for each method. Nowhere do we say that it is possible to always achieve this much reduction. The experiments then go on to perform these calculations for the situations we have actually tested and establish the level of efficiency gain that we can successfully achieve. Thus, we do not see how our statements are loose in any sense. On the point of theoretical analysis, please see our answer to AnonReviewer3.
>
>
> Framework development:
> There is an overarching idea behind our framework, and it is that the performance has dependencies on the parameters, and this can be used to inform decisions on which parameters to train on. We have justified that active learning is the tool to use for this purpose, and also the way it is used. Although we have not studied new objectives (which we also think does not contribute towards our objectives), we have still provided a concrete realization of the framework using an existing one (CVaR) which leads to a sensible algorithm. All these make “not principled” an unjust criticism.
>
>
> Experiments:
> Our experiments successfully establish the robustness and performance of the learned policies and validate the reduction in data usage. Further, we show evidence that the bandit active learner indeed works as it should and plays its part correctly. Quite clearly, this array of experiments is not preliminary.

---

### Official Review · AnonReviewer3 · 2018-11-05
**This paper tries to tackle the sampling efficiency of RL with building a probabilistic surrogate model.**

**Rating:** 6
**Confidence:** 3

**Review:**

This paper targets at a particular type of robust policy search, where a simulation environment exists with explicit tuning parameters, which is referred to as the model parameters of a Markov decision process. The task of robust policy search is to learn a policy robust to all the parameters of the simulator, so that it can potentially give robust performance in real environment. The previous work handles this problem by sampling many trajectories and only learning from the trajectories, in which the current policy produces the worst performance. This approach effectively focus the policy search on the worst case performance, but is highly inefficient as most of the sampled trajectories are discarded. This paper proposes to improve the sampling efficiency by building a surrogate model predicting the return of the current policy given a MDP parameter. The surrogate model is used to select the MDP parameters leading to the worsts performance, so that the policy search can directly sample and learn from the selected MDP parameters without discarding any trajectories.

This paper tries to tackle the sampling efficiency of RL with building a probabilistic surrogate model. This is a promising direction. The biggest concern is that this paper tackles this problem with a combination with existing techniques, leaving many questions unanswered. Presenting the paper in a more theoretical-grounded perspective would make the paper much stronger.

This paper uses a linear stochastic bandits (LSB) method to build a surrogate model of the return of the current policy and fits the surrogate model into the EPOpt framework by sampling from the worst performing parameters according to the surrogate model. As the Thompson sampling algorithm of LSB draw samples from the distribution of MDP parameters that leads to the wost performance, why not directly use it for policy search?

The surrogate model is expected not to give accurate prediction everywhere due to the limited number of data but produces uncertainty of its prediction as an dictator. However, the uncertainty of prediction is not used by the proposed algorithm.

The presentation of the experiment section needs to be improved. The performance of the baseline needs to be explicitly presented, otherwise it is hard to compare. The proposed method will not outperform if the number of trajectories used for updating policy is the same, as the surrogate model can never be as good as the real model. It would be nice to explicitly demonstrate the runtime and performance trade-off.

Minor issues:
1. What does TRPO stand for?
2. When referring to the paper instead of the authors, the citation format needs to be (authors year) instead of authors (year).

---

> ### Author Response · Authors · 2018-11-21
> **Response to review**
>
> Thank you for your review. We have made several changes to the paper as we have described in our official comment. Our response is based on this new version.
>
> > The biggest concern is that this paper tackles this problem with a combination with existing techniques, leaving many questions unanswered.
>
> We request elaboration on what questions are unanswered, and why combining existing techniques is particularly a problem.
>
> > more theoretical-grounded perspective would make the paper much stronger.
>
> Providing theoretical grounding for the effectiveness of our framework in general would be as difficult as doing so for Active Learning itself, that is quantifying how efficiently the Active learner can learn about the output (policy performance) across its input space (the space of model parameters). There hasn’t been a lot of work on providing performance bounds for Active Learning in general (ref. the survey by Settles, 2010). There is however a good amount of analysis that has gone into the specific case we have implemented, namely Linear Bandits with Thompson Sampling. Pl. see (Abeille & Lazaric (2017)) which we have cited for bounds on the deviation of the bandit’s learned weights from their true values.
>
> > As the Thompson sampling algorithm of LSB draw samples from the distribution of MDP parameters that leads to the wost performance, why not directly use it for policy search?
>
> In EffAcTS-EPOpt, the idea is to use the LSB to come up with an approximate sample from the bottom epsilon percentile of parameters according to the source distribution (for the CVaR objective). However, the trajectories generated while the LSB is learning cannot be used for this for two reasons: first, it would have chosen a number of parameters from outside the required region while exploring, and second, it moves towards the absolute worst performance (and so is biased towards that value, even though it learns well about the entire parameter space). That is why we have to have an additional stage of sampling using the performance function fit by the LSB. We illustrate this in Appendix A2 that we have added.
>
> > However, the uncertainty of prediction is not used by the proposed algorithm.
>
> The approach we have taken in our implementation of the framework is to assign the task of uncertainty reduction to the active learner itself (here the LSB), and assume that the learned return profile has low enough uncertainty that we can use it directly for the subsequent steps. We concur that this is a very valid point and that it might be necessary to employ methods that use the uncertainty of predictions to improve the quality of model parameter selection in situations such as very high dimensional model ensembles.
>
> > The proposed method will not outperform if the number of trajectories used for updating policy is the same, as the surrogate model can never be as good as the real model. It would be nice to explicitly demonstrate the runtime and performance trade-off.
>
> A point to be noted is that even EPOpt does not have the true performance profile and also performs approximate sampling from the bottom epsilon percentile (i.e it is also not guaranteed to produce samples from that region alone). Therefore there is no reason to say that this method will not outperform EPOpt, and as we have shown, the performance of EffAcTS-EPOpt is very close to or better than EPOpt. It samples far fewer trajectories to attain this, and so quite clearly wins in the run-time performance tradeoff.

---

### Official Review · AnonReviewer2 · 2018-11-08
**Introducing active learning to robust policy search for efficient sampling.**

**Rating:** 5
**Confidence:** 3

**Review:**

This paper introduced an active learning mechanism on top of robust policy search in RL for better sampling efficiency. The authors proposed EffAcTS active learning framework and combined it with policy search method EPOpt. Theoretical analysis of active learning efficiency was not investigated. Simulation experiments were done on Hopper and Half Cheetah, 5 runs for each parameter setting.

The paper is well written and easy to follow. The authors quickly went through several key topics (active learning, linear bandits, multi-task, etc.) without too many details. However, there is a huge lack of key references in these topics. It would be better to notice that they were not introduced together with DRL.

Overall, it is a nice paper with incremental contributions on every dimension the authors claimed (e.g. comparing to Sharma et al., 2018).

---

> ### Author Response · Authors · 2018-11-21
> **Response to review**
>
> Thank you for your review. We have made several changes to the paper as we have described in our official comment. Our response is based on this new version.
>
> On the point of theoretical analysis, please see our answer to AnonReviewer3.
>
> Due to space constraints, we have discussed only the concepts that are immediately relevant to the contributions of the paper. Going into details such as Deep RL algorithms like TRPO would take up too much space.
>
> We also believe our references on the topics we discuss are exhaustive enough. The survey by Settles (2010) covers work on active learning extensively. We have cited the works that define two kinds of Linear Bandit Algorithms in Section 3.3. Further, we cite several works on bandits that specifically deal with active learning in Section 4. In Section 5, we refer to many prominent MTL papers that are relevant to our discussions apart from (Sharma et. al., 2018). Also see our response to AnonReviewer1 on why approaches like “Model-Agnostic Meta-Learning” are not relevant.
>
> We also have very good reasons to think our paper is not incremental in all dimensions:
>
> * First and foremost, we have achieved a huge reduction in data usage from previous work on Robust RL and experimentally verified the soundness of the framework.
>
> * We have developed a novel framework to use active learning for the problem of Robust Policy Search and provided a concrete realization of the framework which leads to a sensible algorithm (EffAcTS-EPOpt). We have justified the use of active learning in this setting, and brought out the need for developments on that front.
>
> * In our discussions on MTL, we not only compare and contrast with (Sharma et al., 2018), we also clearly bring out how that area of research is tied to this problem through parameterized MTL problems, and how Robust RL can benefit from the application of MTL methods.

---

### Author Response · Authors · 2018-11-21
**Updates on the paper**

We have revised our paper with the following changes:

* Reorganized parts of the description of the EffAcTS framework (section 4) for clarity. We believe that this has been a source of confusion and we make these edits to better express the way in which we use Active Learning in EffAcTS, as well as the usage of the bandit in EffAcTS-EPOpt.

* Added an appendix “Visualizing the Bandit Learner in EffAcTS-EPOpt” where we visualize and  comment on the parameter choices of the bandit algorithm in the learning phase and when outputting trajectories.

* Added a discussion of work on model based RL ([1] and [2]) in the related work (Section 2).

* Errata:
  - Expanded TRPO (Trust Region Policy Optimization) and added inline citations.
  - Changed argument of GetTrajectory (in LEARN, Algorithm 1) to \theta_i, from \pi_\theta_i for consistency with the
    definition of GetTrajectory.
  - Added some references that were missed earlier.
  - Corrected citation formatting throughout.

We agree that presenting the results for EPOpt alongside ours will improve the presentation, and therefore will add it in the camera ready version. As we have pointed out in the paper, we are using the same environments and therefore the results presented in the EPOpt paper can be used for comparison in the meantime.

--------------
[1] Thanard Kurutach,  Ignasi Clavera,  Yan Duan,  Aviv Tamar,  and Pieter Abbeel.   Model-ensemble trust-region policy optimization. In International Conference on Learning Representations, 2018.

[2] I.  Mordatch,  K.  Lowrey,  and  E.  Todorov.    Ensemble-cio:   Full-body  dynamic  motion  planning that transfers to physical humanoids.  In 2015 IEEE/RSJ International Conference on Intelligent Robots and Systems (IROS)

---

### Meta-Review · Area_Chair1 · 2018-12-14
**The paper can be improved**

**Confidence:** 4
**Recommendation:** Reject

**Metareview:**

The paper addresses sample-efficient robust policy search borrowing ideas from active learning. The reviews raised important concerns regarding (1) the complexity of the proposed technique, which combines many separate pieces and (2) the significance of experimental results. The empirical setup adopted is not standard in RL, and a clear comparison against EPOpt is lacking. I appreciate the changes made to address the comment, and I encourage the authors to continue improving the paper by simplifying the model and including a few baseline comparisons in the experiments.